# A simple method of hiPSCs differentiation into insulin-producing cells is improved with vitamin C and RepSox

Ayumi Horikawa[☉], Keiko Mizuno[☉], Kyoko Tsuda, Takayoshi Yamamoto[iD], Tatsuo Michiue[iD]*

Department of Life Sciences (Biology), Graduate School of Arts and Sciences, The University of Tokyo, Tokyo, Japan

[☉] These authors contributed equally to this work.

* tmichiue@bio.c.u-tokyo.ac.jp

**Data Availability Statement:** All relevant data are within the manuscript and its Supporting Information files.

**Funding:** Research Center Network for Realization of Regenerative Medicine program of Japan

## Abstract

Human induced pluripotent stem cells (hiPSCs) are considered a promising source of pancreatic β-cells for the treatment of diabetes. However, this approach is limited by issues such as low efficiency and high cost. Here, we have developed a new protocol to induce insulin-producing cells. To reduce costs, we decreased the number of reagents and replaced protein reagents with chemical compounds. In this method, we increased induction efficiency with ascorbic acid (vitamin C) and an ALK5 inhibitor, RepSox. In 2D culture, the majority of cells were immature β-cells with low glucose-stimulated insulin secretion. Transferring to 3D culture immediately after endocrine progenitor cell differentiation, however, improved glucose-stimulated insulin secretion. This simplified method will contribute to realizing transplantation therapy of β-cells using iPSCs.

## Introduction

Type 1 diabetes results from the destruction of pancreatic β-cells by an autoimmune process resulting in patients having little endogenous control of their blood glucose levels. Currently, the main therapy is through the injection of insulin. Yet, even with insulin injections, blood glucose levels are difficult to properly control, and patients are under high stress due to the life-long need for injections. One of the best options to overcome these issues is transplantation of pancreatic islet cells, including β-cells. This approach is limited by an insufficient number of donors. Moreover, even if the transplantation is successful, immunosuppressants must be taken for the rest of the patient's life [1, 2]. An alternative way is to differentiate pancreatic β-cells from pluripotent stem cells. In particular, many studies have recently been using iPSCs rather than ES cells [3–6], as using iPSCs from the patient negates the issue of both limited donors and transplant rejection.

Pancreas development *in vivo* proceeds as follows: after fertilization, a part of the inner cell mass differentiates into the definitive endoderm, which expresses a homeobox gene, *Sox17* [7]. This then differentiates into embryonic foregut epithelial cells, and starts expressing a homeobox gene, *Pdx1*. Cells then differentiate into pancreatic endocrine progenitor cells,

agency for Medical Research and Development (16bm0304005h0004 to T.M.), MEXT/JSPS KAKENHI (19K16138 to T.Y.).

**Competing interests:** The authors have declared that no competing interests exist.

which express *Nkx6.1*. After a while, the expression of *Neurogenin 3* (*Ngn3*) starts to increase, and cells differentiate into insulin-secreting pancreatic β-cells. Various secreted factors, such as Wnt and FGF, are known to function during these processes [8, 9].

Based on the process *in vivo*, a multistep differentiation protocol was developed to induce pancreatic β-cells from iPSCs. Since originally reported by Baetge *et al*., various modified methods have been published [10–14]. All of these methods, however, are costly due to requiring many chemical compounds and protein reagents. These multistep differentiation protocols are also laborious. Routine use of pluripotent stem cells in regenerative medicine will require a more cost-effective and labor-saving approach.

*In vivo*, when blood glucose levels increase, glucose is incorporated into pancreatic β-cells, and the amount of ATP increases. This closes $K^+$ channels, which depolarizes the cell membrane. This opens $Ca^{2+}$ channels, allowing $Ca^{2+}$ entry into the cell, and insulin secretory granules released. It is still considered to be difficult to induce insulin-secreting, glucose-responsive β-cells *in vitro*. However, it was recently shown that functional β-cells that secrete insulin in a glucose-dependent manner can be efficiently induced in three-dimensional (3D) culture [13, 14].

We previously succeeded in inducing the definitive endoderm from iPSCs after replacing protein reagents with chemical compounds [15]. In the present study, we adapted this method to establish a differentiation protocol to generate pancreatic β-cells with only seven reagents and three steps. Although we could induce insulin-producing cells, they co-expressed glucagon, which should be secreted from pancreatic α cells, and thus the secretion of insulin was insufficient. To improve differentiation efficiency, we added vitamin C and an ALK5 inhibitor, RepSox, into the differentiation medium, and succeeded in suppressing glucagon production and improving insulin production. In addition, glucose-dependent secretion of insulin was improved by 3D culture. Being cost-efficient and with a few steps, our protocol has the potential to significantly contribute to pancreatic β-cell induction technology from iPSCs.

## Materials and methods

### Cell culture and differentiation

The ethical committee of our university authorized this project (University of Tokyo (16–181)). The 201B7 and TKDN4m lines of hiPSCs established from human dermal fibroblasts cells was used in this study [16, 17]. Freeze-stored hiPSCs were thawed and cultured on 6-cm dishes coated with Matrigel (Corning) in serum-free mTeSR1 or mTeSR Plus medium (STEM-CELL Technologies), without mouse embryonic fibroblast feeder cells, at 37˚C under 5% $CO_2$ in air. hiPSCs were cultured with mTeSR1 or mTeSR Plus medium with 10 μM Rho-associated kinase inhibitor (Y-27632; Adooq Bioscience) and passaged at 1:20–1:50 when hiPSCs reached about 80% confluency using 0.02% EDTA (ethylenediaminetetraacetic acid)-PBS. Cell lines used for experiments had less than ten passages in mTeSR1 or mTeSR Plus medium on Matrigel-coated well.

For definitive endoderm (DE) differentiation (stage 1), hiPSCs were dissociated with accutase (Gibco) and plated at a density of 2.8 x $10^4$ cells/$cm^2$ on Matrigel-coated 4-well chamber slides (Thermo Scientific) or 24-well plates with mTeSR1 or mTeSR Plus medium containing 10 μM Y-27632. After one day, the cells were cultured in DIF medium 1 or mDMEM medium (detailed below) with 3 μM CHIR99021 (Wako) and 100 ng/mL Activin A (Activin), with or without 0.25 mM vitamin C (Wako), for 24 h. Undifferentiated cells were cultured in mTeSR1 or mTeSR Plus medium for three days, then cultured in DIF medium 1 or mDMEM medium with 100 ng/mL Ac, with or without 0.25 mM ascorbic acid, for 48 h. At stage 2, cells were cultured for 24 h in DIF medium 2 or mDMEM medium with 1% B27 supplement (B27,

Invitrogen), 10 μM SB431542 (SB, Wako), 0.5 μM LDN193189 (LDN, Sigma), and 2 μM retinoic acid (RA, Wako), with or without 0.25 mM vitamin C. Cells were cultured for five days in DIF medium 2 or modified DMEM/F-12 (Cell Science & Technology Institute) with 1% B27 supplement (B27, Invitrogen), 10 μM SB431542 (SB, Wako), 0.5 μM LDN193189 (LDN, Sigma), and 2 μM retinoic acid (RA, Wako), with or without 0.25 mM vitamin C. At stage 3, cells were cultured for six days in DIF medium 2 or modified DMEM/F-12 with 1% B27, 10 μM SB, and 10 nM GLP-1 (Sigma), with or without 5 μM RepSox (Sigma-Aldrich). At stage 4, cells were cultured for six days in DIF medium 2 or modified DMEM/F-12 with 1% B27, 10 μM SB, and 10 nM GLP-1. Media were changed daily.

Differentiation medium used were as follows:

DIF medium 1: hESF-DIF medium (Cell Science & Technology Institute) containing 10 μg/mL Insulin (Wako), 5 μg/mL Transferrin (Sigma), 500 μg/mL Bovine Serum Albumin (BSA, Sigma), 10 μM sodium selenite, 10 μM ethanolamine (Sigma), and 10 μM 2-mercaptoethanol (Sigma).

DIF medium 2: hESF-DIF containing 5 μg/mL Transferrin (Sigma), 500 μg/mL BSA, 10 μM sodium selenite, 10 μM ethanolamine, 10 μM 2-mercaptoethanol, and 7.5 ng/mL Insulin-like growth factor (IGF-1, Sigma). mDMEM medium: modified DMEM/F-12 containing 10 μM Y-27632.

## Three-dimensional (3D) culture

Two-dimensionally cultured cells were detached from the plate with accutase at the end of stage 3. Cells were then plated at a density of $3 \times 10^5$ cells/mL on 6-well plates with an Ultra-Low Attachment Surface (Corning) with shaking at 95 rpm in 3 mL/well DIF medium 2 containing 10 μM Y-27632.

## Immunohistochemistry

Cells were fixed and immunostained with a standard protocol [15]. Antibodies used were: rabbit anti-Oct4 antibody (1/500; Santa Cruz; sc-9081), goat anti-Sox17 antibody (1/300; R&D; AF1924), mouse anti-Nkx6.1 (1/200; Developmental Studies Hybridoma Bank; F55A10), goat anti-Pdx1 (1/300; R&D; AF2419), mouse anti-Ngn3 (1/300; Developmental Studies Hybridoma Bank; F25A1B3), mouse anti-Insulin (1/600; Sigma-Aldrich; I2018), mouse anti-Glucagon (1/600; Sigma-Aldrich; G2654), rat anti-C-peptide (1/600; Developmental Studies Hybridoma Bank; GN-ID4), anti-rabbit IgG, Alexa Fluor 488 conjugated (1/600; Invitrogen; A21206), anti-goat IgG, Alexa Fluor 594 conjugated (1/600; Invitrogen; A11058), anti-mouse IgG, Alexa Fluor 488 conjugated (1/600; Invitrogen; A21202), anti-rat IgG, Alexa Fluor 594 conjugated (1/600; Invitrogen; A-11007). Nuclei were counterstained with DAPI (1/200; Dojindo; 340–07971) for 2D cultured cells and with TO-PRO3 (Thermo Fisher Scientific) for 3D cultured cells before specimens were mounted in Prolong Gold Antifade Reagent (Invitrogen). Immunostained specimens were examined with an inverted fluorescent microscope (Keyence, Japan) and confocal laser scanning microscope (OLYMPUS, Japan).

## Glucose-stimulated C-peptide secretion assays

Differentiated cells at the end of stage 4 were pre-incubated at 37˚C for 1 h with DMEM (no glucose) containing 10 mM HEPES and 0.1% BSA (normal medium). Cells were washed twice with the normal medium and then incubated at 37˚C for 1 hour. And then, cells were washed twice with the normal medium, incubated for 1 hour in 2.5 mM glucose, washed twice and incubated for 1 hour in 25 mM glucose. This sequential treatment was repeated twice. Then, cells were washed and incubated for 1 hour in 2.5 mM glucose containing 30 mM KCl in

normal medium (400 μL). C-peptide concentrations in the cultured medium were measured using a human C-peptide ELISA kit (Mercodia, Uppsala, Sweden) according to the manufacturer's instructions.

## Statistical analyses

Data are expressed as the mean ± the standard deviation. For comparisons of discrete data sets, paired $t$-test and Welch's $t$-test were used as indicated in the legends. Two-tailed $p<0.05$ was considered significant.

## qRT-PCR

Total RNA was prepared using the RNeasy Plus micro kit (Quiagen). Reverse transcription was carried out using SuperScript III Reverse Transcriptase (Invitrogen). PCR was performed using KOD SYBR qPCR mix (TOYOBO). Primer sets used for qRT-PCR are as follows:

| Gene | Forward Sequence | Reverse Sequence |
|------|------------------|------------------|
| GAPDH | GACATCAAGAAGGTGGTGAA | TGTCATACCAGGAAATGAGC |
| Oct4 | CGAAAGAGAAAGCGAACCAGT | AACCACACTCGGACCACATCC |
| Pdx1 | TCCACCTTGGGACCTGTTTAGAG | CGAGTAAGAATGGCTTTATGGCAG |
| Glucagon | CAGACCAAAATCACTGACAGGAAATA | ACATCCCACGTGGCTAGCA |
| Insulin | AGCCTTTGTGAACCAACACC | GCTGGTAGAGGGAGCAGATG |

Results shown are representative of at least three biological replicates.

## Calcium Imaging

Differentiated cells were dissociated with accutase (Gibco) and incubated with Fluo-8 (AAT Bioquest) for 45 min, and then washed for 15 min in 37°C incubator on a glass base dish. The specimens were visualized by a confocal microscope (FV-1200, Olympus). The images were obtained every 40 second (10 times) after glucose or KCl addition (low glucose, 2.5 mM; high glucose, 25 mM). The proportion of the fluorescence intensity of the cells were normalized by all cell area (bright field image).

## Flow cytometry

The cells were dissociated by Trypsin in 37°C incubator, incubated with FcR Blocking Reagent (Miltenyi) in staining buffer (BD) and fixed with 4% PFA. The specimens were incubated with primary antibodies (mouse anti-Glucagon (Sigma-Aldrich; G2654), rat anti-C-peptide (Developmental Studies Hybridoma Bank; GN-ID4)), and washed by Perm/Wash buffer (BD). The secondary antibodies were anti-mouse IgG, Alexa Fluor 488 conjugated (Invitrogen; A11029), anti-rat IgG, Alexa Fluor 647 conjugated (Life technologies; A-21247). The cells were washed and analyzed using BD FACSCanto™ II Cell Analyzer.

## Results

### A novel and simple method to differentiate iPSCs into pancreatic β-cells

We previously established an efficient and low-cost protocol to induce the definitive endoderm (DE) from iPSCs through the addition of CHIR-99021 (CHIR; an Wnt-signal agonist, GSK3 inhibitor) and Activin A (Ac) [15]. Here, we applied this endoderm differentiation protocol

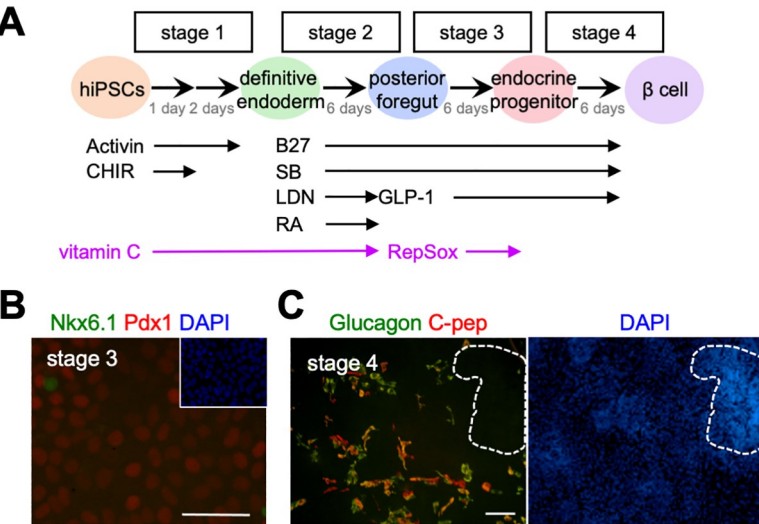

**Fig 1. Analysis of differentiation state of iPSCs induced by a simplified induction method into pancreatic β-cells.**
(**A**) Summary of the protocol. Activin, Activin A; CHIR, CHIR99021; SB, SB431542; LDN, LDN193189; RA, retinoic acid; GLP-1, Glucagon-like peptide-1; RepSox, ALK5 inhibitor. (**B**) Immunohistochemistry at stage 3. The expression of Nkx6.1 (green), Pdx1 (red). DAPI staining shown in blue. Scale bar: 100 μm. (**C**) Immunohistochemistry at stage 4. The expression of glucagon (green) and C-peptide (red). DAPI staining shown in blue. Circles with a dashed line indicate the mass region. Scale bar: 100 μm.

for the generation of pancreatic β-cells (outlined in Fig 1A). In this method, chemical compounds replaced protein reagents to reduce costs. Briefly, at stage 1, Ac and CHIR were added to the medium to differentiate the human iPSC line, 201B7 cells, into the definitive endoderm. At stage 2, B27 supplement, SB431542 (SB, an inhibitor of the TGF-β pathway), LDN193189 (LDN, an inhibitor of the BMP pathway), and retinoic acid (RA) were added to the medium. At stage 3 and 4, B27, SB, and Glucagon-like peptide-1 (GLP-1) were added to the medium and incubated for 12 days (Fig 1A).

At stage 1, immunohistochemistry (IHC) showed many cells were positive for Sox17, but negative for Oct4 (an undifferentiated marker; S1A Fig). Similar results were obtained by qRT-PCR as reported (S1B Fig) [15]. At stage 3, Pdx1 expression was detected in many cells, whereas only a few Nkx6.1-positive cells were observed (Fig 1B). At stage 4, C-peptide, which is secreted with insulin, was detected by IHC (Fig 1C). This result indicated insulin-producing cells had been successfully induced.

We observed that the induction was not proceeded similarly across all the cells. When the cell density was high at stage 2, the cells often tended to pile up at stage 3 resulting in the formation of a mass region (Fig 1C, circle with a dashed line). At stage 4, a few C-peptide-positive cells were observed in the mass region, but more C-peptide-positive cells were found in the monolayer region (Fig 1C). The ratio of C-peptide-positive cells in the monolayer region, however, was not high, and many cells also expressed glucagon (Fig 1C), indicating the cells were not well differentiated into mature β-cells. To improve the induction efficiency in this simplified method, we added vitamin C and RepSox into the medium.

## Vitamin C or RepSox can improve the induction efficiency of endocrine progenitor cells

Several methods have been reported to increase the efficiency of differentiation into pancreatic endocrine cells. For example, the addition of vitamin C during the early induction stage has

been reported to improve the efficiency of pancreatic endoderm induction [13]. The inhibition of TGF-β signaling by a chemical compound, RepSox, in the later stages increases the efficiency of endocrine progenitor induction and the expression of insulin [13, 18]. Thus we applied vitamin C and RepSox into our simplified method (Fig 1A, magenta). The addition of vitamin C resulted in a significant decrease in *Oct4* expression compared with no VC (control) (Fig 2A). *Sox17* and *FoxA2* expression levels did not show any clear differences (Fig 2A). IHC confirmed that most cells both in the control and vitamin C-treated cells similarly expressed Sox17 and did not expressed Oct4 (Fig 2B). Close examination, however, indicated there were slightly more cells expressing neither Oct4 nor Sox17 when vitamin C was added compared with the control (Fig 2C, arrowhead). Vitamin C increased the cell number (Fig 2D) and the proportion of cells expressing neither Oct4 nor Sox17 slightly increased with the addition of vitamin C (Fig 2E, Sox17-Oct4-).

Vitamin C was added until stage 2, and RepSox was added from stage 3 (Fig 1A). In these protocols, cells formed a monolayer region and a mass region (surrounded by a dotted line) similar to control without vitamin C and RepSox (Fig 3A). As previously described, the mass region did not well differentiated into β-cells (Fig 1C), so we evaluate especially in a monolayer region. The addition of vitamin C increased Ngn3-positive cells in the monolayer region, and the addition of RepSox increased Nkx6.1-positive cells (Fig 3A and 3B). Almost all cells in the monolayer region expressed Pdx1, but there were fewer Pdx1-positive cells in the mass region (Fig 3A). In more detail. 0.029 mm$^2$ foci from the monolayer region were evaluated and the number of cells positive for Pdx1, Ngn3, Nkx6.1, Pdx1/Ngn3, and Pdx1/Nkx6.1 were counted, along with cell density and the ratio of positive cells (Fig 3C–3F). The addition of vitamin C increased cell density by more than 1.5-fold (Fig 3C). Many cells were Pdx1-positive (about 90%), which increased when vitamin C was added (to more than 95%, Fig 3D). Vitamin C resulted in a 5-fold increase in the percentage of Ngn3-positive cells (30%) compared with the control, while RepSox increased this cell population from 6% to 12% (Fig 3E). The addition of both vitamin C and RepSox did not result in a significant increase in the percentage of Ngn3-positive cells beyond that observed with vitamin C alone (Fig 3E). The addition of vitamin C reduced the number of Nkx6.1-positive cells, whereas the addition of RepSox increased the number of Nkx6.1-positive cells (Fig 3B and 3F). When both vitamin C and RepSox were added, the expression of Nkx6.1 increased more than 3-fold compared with the control (Fig 3F). Differentiation of endocrine progenitor cells to pancreatic β-cells has been reported to require the co-expression of Pdx1 and Nkx6.1 [19, 20]. With vitamin C alone, Pdx1- and Nkx 6.1-positive cells were rarely observed, but with RepSox alone, the percentage of Pdx1– and Nkx6.1-positive cells increased about 2-fold (Fig 3F). When both vitamin C and RepSox were added, there was a significant increase in the number of cells expressing both Pdx1 and Nkx6.1 about 10-fold (Fig 3F).

The expression of the pancreatic differentiation marker, *Pdx1*, was also quantified by qRT-PCR. The addition of vitamin C increased *Pdx1* expression 5-fold compared with the control (Fig 3G). Given there was no increase of the percentage of cells expressing Pdx1 in the IHC result (Fig 3D), it suggests vitamin C was increasing *Pdx1* expression per cell. Consistent with this, IHC showed the intensity of Pdx1 per cell appeared slightly higher in the vitamin C group (Fig 3A and 3B). These results indicate the addition of vitamin C and/or RepSox significantly increases the percentage of Ngn3-positive and Nkx6.1-positive cells, as well as the expression of *Pdx1* within cells.

## Vitamin C significantly increases insulin-producing cells

The addition of vitamin C clearly increased the number of insulin-producing cells and Pdx1-positive cells compared with the control (Fig 4A). Consistent with this, the proportion of

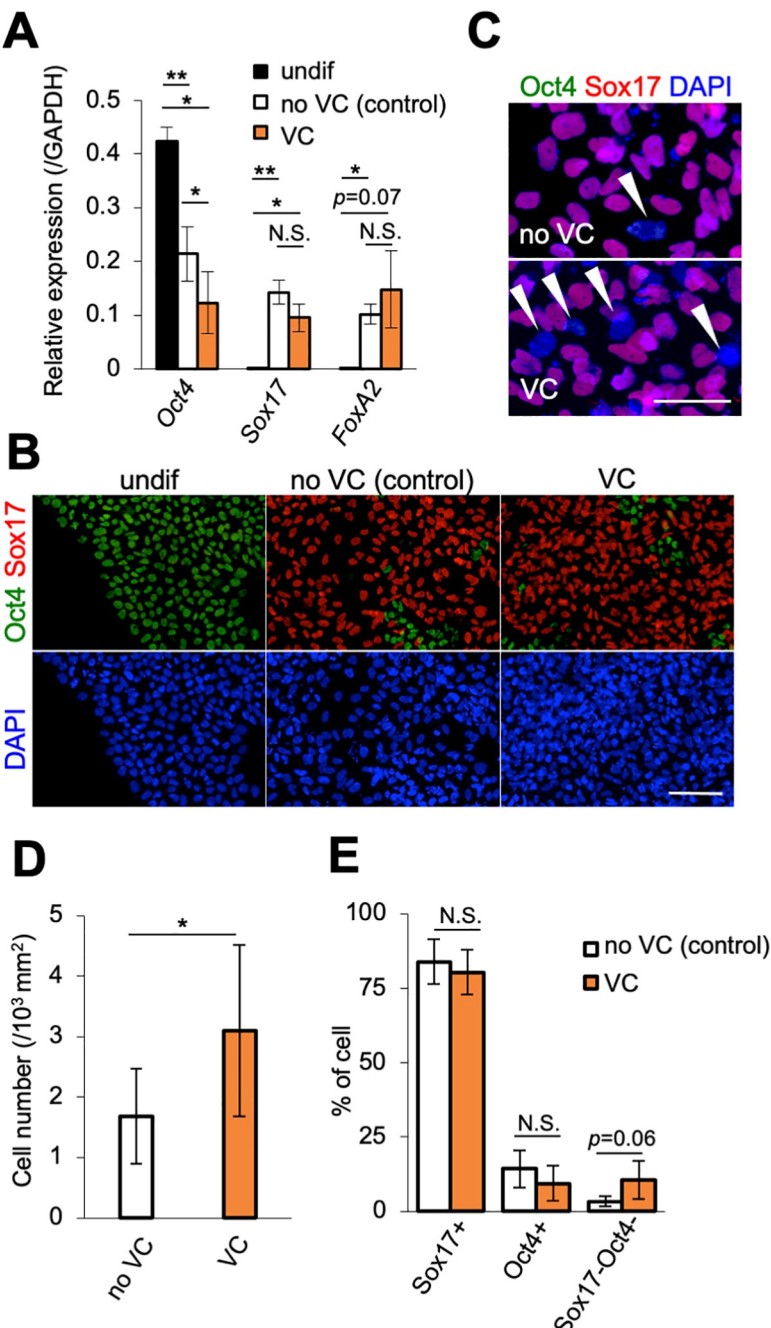

**Fig 2. Analysis of differentiation into endoderm at stage 1.** (**A**) qRT-PCR analysis of the expression of *Oct4*, *Sox17* and *FoxA2*, normalized against *GAPDH* at stage 1. undif (undifferentiated cells), black; no VC (control), white; VC (vitamin C-treated), orange. Horizontal bars indicate statistical analyses. $*p < 0.05$, $**p < 0.01$, N.S. not significant; paired *t*-test. (**B, C**) IHC for Oct4 (green) and Sox17 (red) at stage 1. A higher magnification of no VC (control) and VC (vitamin C-treated) cells was shown in (**C**). Arrowheads in C indicate cells expressing neither Oct4 nor Sox17 (blue cells). Nuclei were stained with DAPI (blue). Scale bars: 100 μm (**B**), 50 μm (**C**). (**D**) The number of cells in 1 mm$^2$ foci from the monolayer region. no VC, white; VC (vitamin C-treated), orange (n = 6, including two biological and three technical replicates). $*p < 0.05$; paired *t*-test. (**E**) Percentage of cells expressing Sox17 (Sox17+), Oct4 (Oct4+) and neither Sox17 nor Oct4 (Sox17-/Oct4-). no VC (control), white; VC (vitamin C-treated), orange (n = 6, including two biological and three technical replicates). Statistical analysis was performed by paired *t*-test.

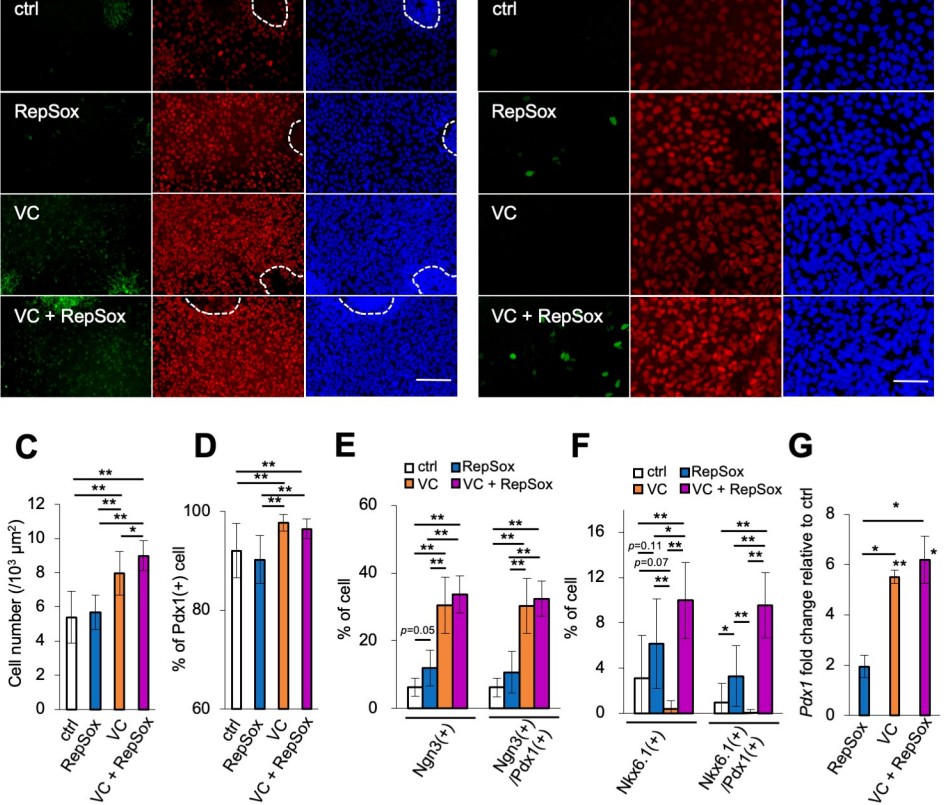

**Fig 3. Addition of vitamin C or RepSox increased the rate of differentiation into pancreatic progenitor cells (stage 3).** (**A, B**) IHC for Ngn3 (green) and Pdx1 (red) (**A**) and for Nkx6.1 (green) and Pdx1 (red) (**B**) at stage 3. DAPI staining shown in blue. Circle with a dashed line in **A** shows the mass region. Scale bars: 100 μm (**A**), 50 μm (**B**). (**C**) Cell number in the monolayer region, measured by DAPI-stained images: control, white; RepSox, blue; VC (vitamin C-treated), orange; VC (vitamin C-treated) + RepSox, magenta (n = 15, including five biological and three technical replicates). $^*p<0.05$, $^{**}p<0.01$; Welch's $t$-test. (**D-F**) Percentage of cells expressing the indicated differentiation markers in 0.029 mm² foci from the monolayer region using IHC results: control (no VC and no RepSox), white; RepSox, blue; VC, orange; VC + RepSox, magenta. Percentage of Pdx1(+) cells (n = 15, including five biological and three technical replicates) (**D**). Percentage of cells of Ngn3(+) and Ngn3(+)/Pdx1(+) (n = 6, including two biological and three technical replicates) (**E**). Percentage of cells of Nkx6.1(+) and Nkx6.1(+)/Pdx1(+) (n = 9, including three biological and three technical replicates) (**F**). Horizontal bars indicate statistical analysis in **E, F**. $^*p<0.05$, $^{**}p<0.01$; Welch's $t$-test. (**G**) Fold change expression level of *Pdx1* compared with the control at stage 3: RepSox, blue; VC, orange; VC + RepSox, magenta (three biological replicates per group) by qRT-PCR (normalized against the control (no vitamin C and no RepSox)). $^*p<0.05$, $^{**}p<0.01$; paired $t$-test. The asterisk shows a significant difference compared with the control.

C-peptide-producing cells significantly increased with the addition of vitamin C (Fig 4B and 4C). Conversely, the addition of RepSox, with or without vitamin C, had no significant effect on the number of insulin–and C-peptide-producing cells (Fig 4A–4C). The number of gluca-gon-expressing cells did not significantly change with the addition of vitamin C or RepSox (Fig 4B). The number of Nkx6.1-expressing cells increased with the addition of vitamin C and/or RepSox (Fig 4C). We next roughly estimated the proportion of cells expressing C-peptide and glucagon by measuring the area of C-peptide and glucagon expression. The glucagon expression area increased about 1.5-fold with RepSox, compared with the control, but there was no significant change with vitamin C (S2 Fig). Conversely, the C-peptide expression area did not change with the addition of RepSox, but significantly increased with the addition of

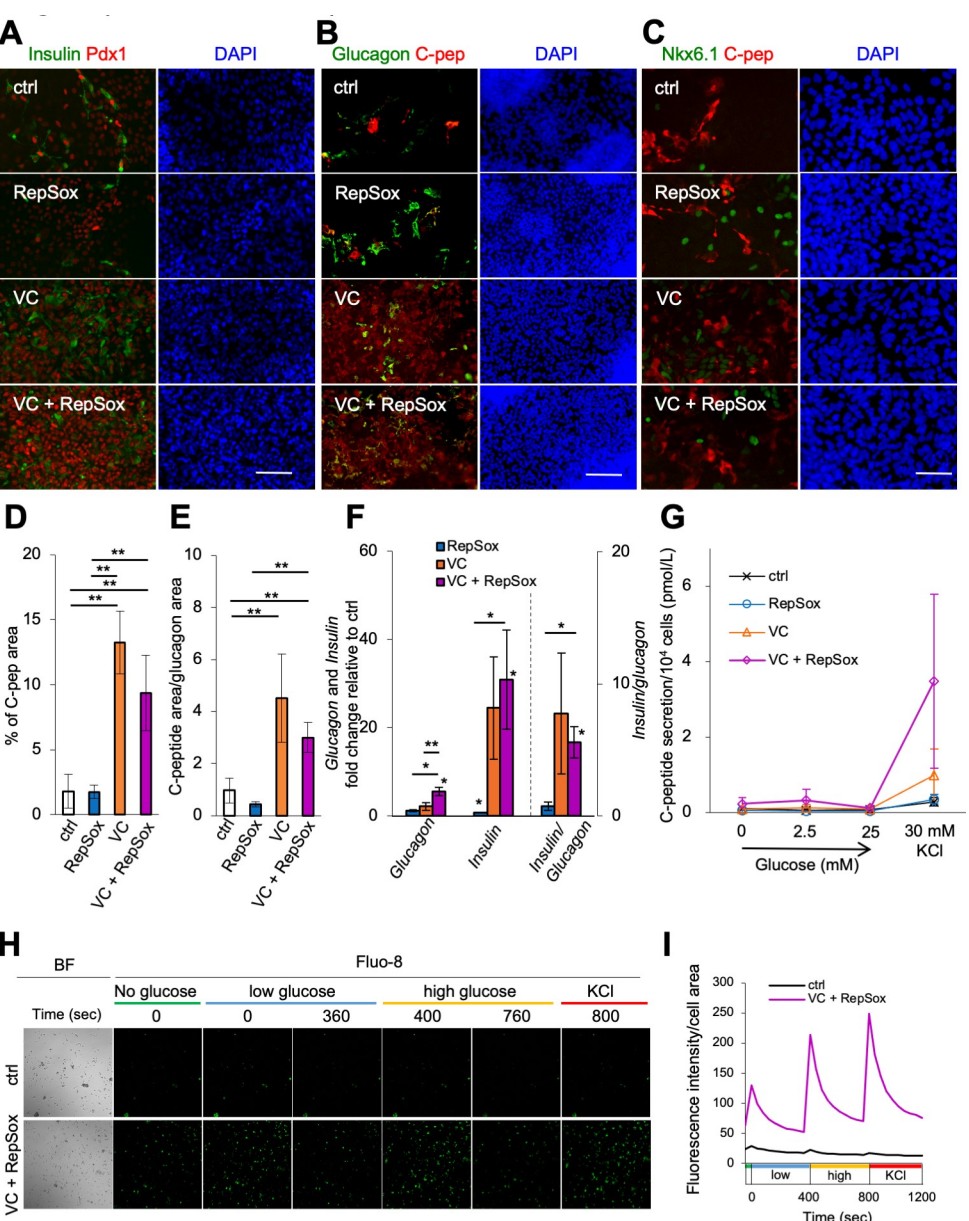

**Fig 4. Vitamin C addition increased the number of insulin-producing cells (stage 4).** (**A-C**) IHC. The expression of insulin (green) and Pdx1 (red) (**A**). The expression of glucagon (green) and C-peptide (red) (**B**). The expression of Nkx6.1 (green) and C-peptide (red) (**C**). DAPI staining shown in blue. Scale bars: 100 μm (**A, B**), 50 μm (**C**). (**D**) The percentage of the cell area expressing C-peptide in the monolayer region, corresponding to **B** (n = 5, including one biological and five technical replicates). $^{*}p<0.05$, $^{**}p<0.01$; Welch's $t$-test. (**E**) The C-peptide-positive cell area divided by glucagon-positive area. Glucagon-positive cell areas in a certain region of the monolayer cells were measured using **B** (n = 5, including one biological and five technical replicates). Control, white; RepSox, blue; VC, orange; VC + RepSox, magenta. $^{*}p<0.05$, $^{**}p<0.01$; Welch's $t$-test. (**F**) Fold change of the expression levels of the indicated genes and *insulin/glucagon* compared with the control at stage 4 by qRT-PCR (n = 3 biological replicates per group). RepSox, blue; VC, orange; VC + RepSox, magenta. Asterisks represent statistical significance. $^{*}p<0.05$, $^{**}p<0.01$; paired $t$-test. The asterisk on the bar graph shows a significant difference compared with the control. (**G**) The amount of C-peptide secretion determined by ELISA. Control, black; RepSox, blue; VC, orange; VC + RepSox, magenta. (**H**) Fluorescence image of Fluo-8 staining cells. Ctrl and VC + RepSox treated cells challenged with no glucose, low (2.5 mM) and high (25 mM) glucose and 30 mM KCl. (**I**) Line graphs of the time-course of Ca2+ flux level in control (blue) and VC + RS (orange) treated cells during glucose or KCl challenge.

vitamin C (Fig 4D). Accordingly, the C-peptide/glucagon ratio area significantly increased with the addition of vitamin C (Fig 4E).

To confirm these results, the expression of *insulin* and *glucagon* were measured by qRT-PCR. Consistent with IHC, when vitamin C was added, the expression of *insulin* slightly increased (Fig 4F). The expression of *glucagon* also increased, but at a lower rate than that of *insulin*, and the ratio of *insulin*/*glucagon* greatly increased with the addition of vitamin C (Fig 4F). Taken together, these results suggest the addition of vitamin C promotes differentiation into insulin-producing cells, which do not co-express glucagon. To determine whether these cells could respond to glucose stimulation, we measured C-peptide secretion (Fig 4G). While C-peptide was not almost detected with glucose stimulation in the control and RepSox, a small amount of C-peptide was released by KCl stimulation. When vitamin C was added, there was little response to glucose, but about 3-fold increase in the release of C-peptide by KCl stimulation compared with the control (Fig 4G, orange). When both vitamin C and RepSox were added, C-peptide secretion was slightly stimulated by 2.5 mM glucose addition (Fig 4G, magenta). Interestingly, the KCl-dependent release of C-peptide increased more than 10-fold compared with the control, demonstrating vitamin C and RepSox synergistically increase the KCl-dependent release of C-peptide. Taken together, these results indicates that RepSox does not contribute to the expression of C-peptide but has a significant effect on KCl-dependent insulin secretion. To examine calcium flux in response to glucose (low, high) and KCl, we treated a synthetic calcium indicator, fluo-8 AM. Basal level (no glucose addition) of the fluorescence was higher in vitamin C and RepSox added cells, and the fluorescence increased immediately after the addition of glucose or KCl in the control and vitamin C + RepSox treated condition (Fig 4H; see also materials and methods). The change of the signal intensity every 40 second after glucose or KCl addition was measured and visualized as a graph (Fig 4I). This shows the intensity and the change after the addition of glucose or KCl was higher in vitamin C and RepSox added cells.

## Three-dimensional culture improves insulin secretion ability in response to glucose stimulation

The addition of vitamin C and RepSox can increase the generation of insulin-producing cells in our simplified protocol, but we did not observe clear C-peptide secretion in response to glucose. To evaluate our method, we carried out differentiation by another differentiation method [4]. qRT-PCR analysis indicated that *Insulin* expression of the cells by our method was higher than that by the other method, whereas *MafA* expression was lower, suggesting the immaturity of differentiated cells by our method (S3 Fig). It has been reported that β-cell maturation can be improved in 3D culture compared with 2D culture, with gene expression patterns more similar to that observed *in vivo* [4, 21, 22]. As such, we differentiated our cells in 3D culture. At the end of stage 3, cells were dissociated with accutase and transferred to 3D culture with the same differentiation compounds as used for 2D culture. In the presence of vitamin C and RepSox, *Insulin* expression in 3D culture showed tendency to be higher than that in 2D culture (Fig 5A, the middle graph). But *Glucagon* expression also increased, thus the *Insulin*/*Glucagon* expression ratio did not clearly increase by spheroid formation (Fig 5A, the left and right graphs). In 3D culture, vitamin C and RepSox increased the expression of both *insulin* and *glucagon* (Fig 5A). Immunohistochemistry also showed that vitamin C and RepSox increased expression of both insulin and glucagon in spheroids (Fig 5B). Occasionally, spheroids had hollow cysts (S4A Fig, arrowhead). In this case, *insulin* and *glucagon* expression did not increase with vitamin C and RepSox, compared with that in 2D (S4B–S4D Fig, experiment 3).

We then assessed C-peptide release at stage 4. In 3D culture, the addition of vitamin C and RepSox resulted in a 1.5-fold increase in C-peptide secretion in response to KCl, compared

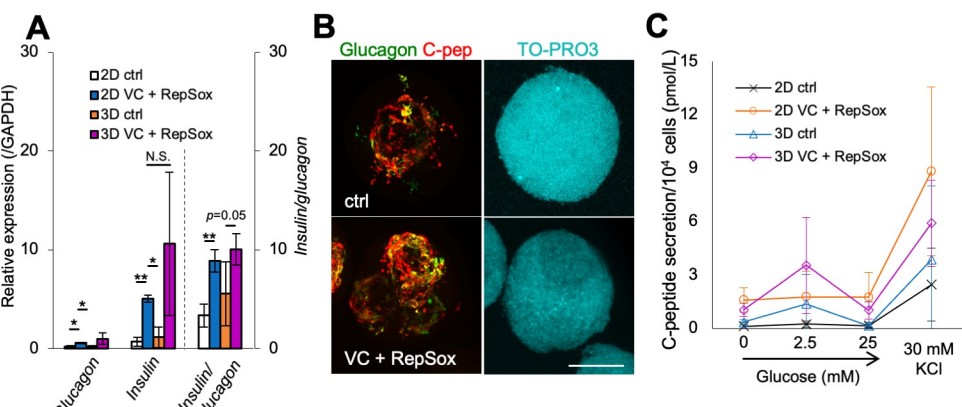

**Fig 5. Insulin responsiveness obtained by 3D culture.** (**A**) qRT-PCR analysis of the expression normalized against *GAPDH* and *insulin/glucagon* at stage 4. 2D control, white; 2D VC + RepSox, orange; 3D control, blue; 3D VC + RepSox, magenta. The data show the average of three experiments (each result was showed in S4B–S4D Fig). *$p < 0.05$, **$p < 0.01$; paired *t*-test. (**B**) IHC of 3D culture cell at stage 4. The expression of glucagon (green) and C-peptide (red). TO-PRO3 staining (DNA) shown in blue. Maximum intensity projection of z-stack images. Scale bar, 100 μm. (**C**) C-peptide secretion determined by ELISA. 2D control, black; 2D VC + RepSox, orange; 3D control, blue; 3D VC (vitamin C-treated) + RepSox, magenta. The data show the average of three experiments.

with 3D control, but this increase was not as pronounced as that observed in 2D culture (Fig 5C, most right). However, 3D cultured cells had acquired the ability to respond to low levels of glucose, with an increase in C-peptide secretion observed in the untreated 3D control, which doubled with the addition of vitamin C and RepSox (Fig 5C). These results indicate 3D culture enhances insulin secretion in response to glucose.

## Discussion

In this study, we establish a cost-efficient and simplified differentiation protocol for pancreatic β-cells. With this simple protocol, the addition of vitamin C and RepSox to the medium improves the induction of insulin-producing cells. We mainly used 201B7 in this study, and we checked that similar induction efficiency was observed in another cell line (S5A-S5C Fig). In addition, we suggest RepSox may contribute to the formation of the secretory process rather than the differentiation to insulin-producing cells (explained as follows).

In general, a variety of signaling factors are used to mimic the differentiation process for β-cell induction. In our method, these factors were replaced with chemical compounds, and the number of steps were reduced. Differentiation into β-cells *in vivo* involves many types of cells and factors. This includes the secretion of inhibitory factors by neighboring cells, which are subsequently suppressed by factors expressed by pancreatic primordia. Conversely, only a few cell types are known to exist in *in vitro* culture. As such, there are not as many factors that act between cells as *in vivo*, and therefore, little or no inhibitors to suppress. This demonstrates it is possible to differentiate β-cells *in vitro* more efficiently with fewer factors and fewer steps than commonly used methods by the addition of vitamin C and RepSox with 3D culture. In our simplified method, the addition of vitamin C, not only increased C-peptide production, but also slightly increased the number of Sox17-negative cells at stage 1. It has been reported that differentiation into pancreatic β-cells is promoted by the presence of other types of cells, such as blood vessels [23–27]. This suggests that Sox17-negative cells may promote differentiation into pancreatic β-cells. Further investigation is required to determine what Sox17-negative cells differentiate into and how they affect β-cell differentiation. In the presence of vitamin C, RepSox addition did not increase insulin production but did increase KCl-stimulated

secretion (Fig 4B–4E and 4G). This suggests RepSox maturates the glucose-response pathway leading to secretion. Under vitaminC and Repsox conditions, significant increase in Ca2+ flux by either not only KCl but also glucose stimulation was observed (Fig 4H and 4I), supporting the increase of secretion activity by Repsox. Previous studies have shown that treatment with an ALK5 inhibitor promotes differentiation of pancreatic α and β-cells [18, 28]. Our study advances this understanding by showing that RepSox may contribute to maturation of the secretory response rather than the differentiation to insulin-producing cells.

Three-dimensional culture improved insulin secretion as reported previously [13, 14]. Differentiated cells are multilayered and polarized *in vivo*. Since cell polarity has been reported to be required for the maturation of pancreatic β-cells [29], it is possible that 3D culture promotes proper cell orientation and glucose responsiveness. In our 3D method, glucose stimulation induced the secretion of C-peptide. Reports indicate these immature β-cells can be matured when transplanted into living organisms or cultured with other cell types [28, 30]. Transplantation experiments will determine if our 3D-cultured cells can be matured with sufficient glucose responsiveness. Another aspect that could be optimized is the timing of when to start 3D culture to improve maturation. Three-dimensional culture occasionally gave rise to spheroids with hollow cysts, which did not well express *insulin*. β-cells induced from iPSCs of a diabetic patient frequently produced spheroids with a hollow cyst that did not differentiate into insulin-producing cells [31], suggesting that a hollow cyst can be an indicator of the failure in β-cell differentiation from iPSCs.

Our method could also be used to understand pancreatic development. Many studies have identified many genes involved in pancreatic differentiation, however, due to their vast number, it has been challenging to identify which factors and pathways are important in pancreatic development. Our simplified method could help elucidate core pathways in pancreatic differentiation.

## Supporting information

**S1 Fig. Definitive endoderm differentiation in this method.** (**A**) Immunohistochemistry of the cells at stage 1. The expression of Oct4 (green) and Sox17 (red). DAPI staining shown in blue. Scale bar: 100 μm. (**B**) qRT-PCR analysis of the expression of *Oct4* (black), *Sox17* (gray), and *FoxA2* (white), normalized against *GAPDH* at stage 1 (three technical replicates per group).
(TIF)

**S2 Fig. The percentage of the cell area expressing glucagon in the monolayer cells.** The area corresponds to Fig 4B (n = 5, including one biological and five technical replicates). control, white; RepSox, blue; VC (vitamin C-treated), orange; VC (vitamin C-treated) + RepSox, magenta. $^*p<0.05$, $^{**}p<0.01$; Welch's paired *t*-test.
(TIF)

**S3 Fig. Comparison of differentiation efficiency between our method and another published method.** (**A**) Immunohistochemistory of the cells. The expression of glucagon (green) and C-peptide (red). DAPI staining shown in blue. Scale bar: 100 μm. (**B**) qRT-PCR analysis of the expression of *Insulin* (black) and *Glucagon* (white). (**C**) qRT-PCR analysis of the expression of *MafA* (gray).
(TIF)

**S4 Fig. The result of each differentiation experiment in Fig 5.** (**A**) The morphology of spheroids in three independent experiments. The arrowhead indicates a hollow cyst. Scale bar: 100 μm. (**B, C, D**) qRT-PCR analysis of the expression of *insulin* (**B**), *glucagon* (**C**) *and insulin/*

*glucagon* (**D**), normalized against *GAPDH* at stage 4, corresponds to Fig 5A. 2D control, white; 2D VC (vitamin C-treated) + RepSox, orange; 3D control, blue; 3D VC (vitamin C-treated) + RepSox, magenta.
(TIF)

**S5 Fig. Validation of our method in another cell line, TKDN4m.** (**A**) Immunohistochemistry of the cells. The expression of glucagon (green) and C-peptide (red). DAPI staining shown in blue. Scale bar: 100 μm. (**B**) qRT-PCR analysis of the expression of *Insulin* (black) and *Glucagon* (white). (**C**) Flow cytometry analysis of C-peptide and glucagon expressing cells.
(TIF)

## Acknowledgments

We greatly acknowledge Dr. Makoto Asashima, Dr. Kiyoshi Ohnuma, Dr. Shuji Takahashi and Dr. Hiromasa Ninomiya for their support. We also thank Ms. Reiko Terada, Mr. Toshiyuki Miura, Ms. Akiko Hara, Ms. Yuna Naraoka for their technical help. We also appreciate Prof. Atsushi Miyajima and Dr. Tohru Itoh for FACS analysis, and Dr. Kazuki Harada for Ca2+ flux assay. The human iPS cell line was kindly provided by RIKEN cell bank and Institute of Medical Sciences, the University of Tokyo. All authors read and approved the manuscript.

## Author Contributions

**Conceptualization:** Tatsuo Michiue.

**Data curation:** Tatsuo Michiue.

**Funding acquisition:** Tatsuo Michiue.

**Investigation:** Ayumi Horikawa, Keiko Mizuno, Kyoko Tsuda, Tatsuo Michiue.

**Project administration:** Tatsuo Michiue.

**Supervision:** Takayoshi Yamamoto, Tatsuo Michiue.

**Validation:** Tatsuo Michiue.

**Writing – original draft:** Ayumi Horikawa, Keiko Mizuno, Tatsuo Michiue.

**Writing – review & editing:** Keiko Mizuno, Kyoko Tsuda, Takayoshi Yamamoto, Tatsuo Michiue.

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
