## [Decision Letter · Decision Letter 0]

27 Apr 2021

PONE-D-21-09954

A simple method of hiPSCs differentiation into insulin-producing cells is improved with vitamin C and RepSox

PLOS ONE

Dear Dr. Michiue,

Thank you for submitting your manuscript to PLOS ONE. After careful consideration, we feel that it has merit but does not fully meet PLOS ONE’s publication criteria as it currently stands. Therefore, we invite you to submit a revised version of the manuscript that addresses the points raised during the review process.

Your manuscript was reviewed by two knowledgeable referees in this area, and their  comments are appended. As you will see they both had several major concerns that will need to be addressed by the authors before I can proceed further. The authors need to properly address/respond to their all comments to fully satisfy our reviewers.

We look forward to receiving your revised manuscript.

Kind regards,

Makoto Kanzaki, Ph.D.

Academic Editor

PLOS ONE

Journal Requirements:

The human iPS cell line was kindly provided by RIKEN cell bank. This study was supported in part by the Research Center Network for Realization of Regenerative Medicine program of Japan agency for Medical Research and Development  (16bm0304005h0004 to T.M.), and MEXT/JSPS KAKENHI (19K16138 to T.Y.). We greatly acknowledge Makoto Asashima, Kiyoshi Ohnuma, Shuji Takahashi, Hiromasa Ninomiya for their support. We also thank Reiko Terada, Toshiyuki Miura, Akiko Hara, Yuna Naraoka for their technical help. All authors read and approved the manuscript.

Research Center Network for Realization of Regenerative Medicine program of Japan agency for Medical Research and Development (16bm0304005h0004 to T.M.), MEXT/JSPS KAKENHI (19K16138 to T.Y.).

3. Please include captions for your Supporting Information files *at the end of your manuscript*, and update any in-text citations to match accordingly. Please see our Supporting Information guidelines for more information: http://journals.plos.org/plosone/s/supporting-information.

Reviewers' comments:

Reviewer's Responses to Questions

**Comments to the Author**

1. Is the manuscript technically sound, and do the data support the conclusions?

Reviewer #1: Yes

Reviewer #2: Partly

2. Has the statistical analysis been performed appropriately and rigorously? 

Reviewer #1: Yes

Reviewer #2: I Don't Know

3. Have the authors made all data underlying the findings in their manuscript fully available?

Reviewer #1: Yes

Reviewer #2: Yes

4. Is the manuscript presented in an intelligible fashion and written in standard English?

Reviewer #1: Yes

Reviewer #2: Yes

5. Review Comments to the Author

Reviewer #1: First of all, I should appreciate the author’s efforts in researching in improving the protocol in the generation of iPS-derived beta cells, which I believe would be one of the sources to cure T1D in near future!

Major concerns:

The authors claimed that they provided a simple and cost-effective protocol to generate iPS-derived beta cells by emphasizing the application of Vit C and ALK-5 inhibitor and 3-D culture. Although the provided results support the idea, the application of the above-mentioned reagent and 3-D culture has been proven already by Rezania et al (Nat. biotech 2014) and Massumi et al (PLOS ONE).

The authors should compare the number of C-peptide positive cells after optimization with the above-mentioned studies to give a clear picture of their protocol vs others.

Although the authors claimed that 3-D culture can increase the maturity of the cells by GSIS doesn't show such a significant difference in Fig5C.

To prove the maturity the authors didn’t show eth Ca2+ efflux in the differentiated cells.

As it has been proven, hES is more efficient in the generation of PS-derived Beta cells, which would be more informative if they test the ES cells alongside iPS using the proposed samples.

Lacking human islets or beta-cell lines making hard to judge the correct performance of the techniques.

Lack of flow cytometry to quantify the percentage of insulin-positive cells.

Minor concerns:

The quality of the pictures is low.

In M&M in part of GSIS, they didn’t mention 25 mM glucose.

The action the authors should take for re-reviewing the paper:

Confirming the percentage of the insulin-producing cells in the population by flow cytometry.

Confirming the maturity of the differentiated cells by measuring the intracellular Ca+2.

Including the ES-derived beta cells as comparing with iPS-derived beta cells

Including the human islets to confirm the accuracy of the technique and comparing with the maturity of the cells

Reviewer #2: The authors reported a method for differentiating insulin-producing cells from hiPSCs and improved glucose-stimulated insulin secretion by adding vitamin C and RepSox as well as by 3D culture.

The weak point of this paper is that the authors use only one cell line as hiPSCs for their experiments. It is difficult to say whether this protocol may be applicable to other hiPSCs unless they show similar data using more than two different cell lines, because hiPSCs have different characteristics.

Although the authors concluded that they could improve glucose-stimulated insulin secretion, I do not think the supporting data are enough to say so. According to the Figure 5C, C-peptide secretion decreased at 25 mM glucose concentration compared with the level at 2.5mM, which indicates the lack of glucose responsiveness. Moreover, the description of glucose-stimulated C-peptide secretion assays in M&M is not accurate.

I agree with the authors about the necessity of cost reduction for cell therapy. Many researchers tried to induce functional islets by imitating the developmental process for pancreatic islets. It is important to show that the quality of induced βcells did not change despite the simplified method. It would be more convincing if the authors present comparison data with other reported methods to say “ This simplified method will contribute to realizing transplantation therapy of βcells using iPSCs.”

6. PLOS authors have the option to publish the peer review history of their article (what does this mean?). If published, this will include your full peer review and any attached files.

Reviewer #1: No

Reviewer #2: No

---

## [Author Response · Author response to Decision Letter 0]

10 Jun 2021

Reviewer #1: First of all, I should appreciate the author’s efforts in researching in improving the protocol in the generation of iPS-derived beta cells, which I believe would be one of the sources to cure T1D in near future!

Major concerns:

The authors claimed that they provided a simple and cost-effective protocol to generate iPS-derived beta cells by emphasizing the application of Vit C and ALK-5 inhibitor and 3-D culture. Although the provided results support the idea, the application of the above-mentioned reagent and 3-D culture has been proven already by Rezania et al (Nat. biotech 2014) and Massumi et al (PLOS ONE).

The authors should compare the number of C-peptide positive cells after optimization with the above-mentioned studies to give a clear picture of their protocol vs others.

 We appreciate all the reviewer’s comments. We have not directly compared the differentiation efficiency with other protocols before, so we compared with one of other methods (Yabe et al., 2017). As a result, the ratio of insulin positive cells (IHC) as well as insulin expression (qRT-PCR) in differentiated cells by our method were larger than “another” method, whereas Maf-A expression in our differentiated cells was lower than another method. Nevertheless, this result indicates that our method is basically comparable with another method. The result of the experiment was shown in S3 Fig.

Although the authors claimed that 3-D culture can increase the maturity of the cells by GSIS doesn't show such a significant difference in Fig5C. To prove the maturity the authors didn’t show the Ca2+ efflux in the differentiated cells.

 Thank you for the comment. Our result indicated that insulin secretion was increased in at least 2.5 mM glucose challenge. As the reviewer points out, 25 mM challenge did not increase insulin secretion. 

To evaluate the maturity of our differentiated cells, we examined Ca2+ flux in differentiated cells using Fluo-8. As a result, increase of Ca2+ flux by glucose challenge as well as KCl treatment was observed in VC + RS condition. This result is newly shown in Fig. 4H and 4I (and mentioned in main text, P13L23-30). 

As it has been proven, hES is more efficient in the generation of PS-derived Beta cells, which would be more informative if they test the ES cells alongside iPS using the proposed samples.

Thank you for the comment. We understand the importance of evaluating our method using hESC. Unfortunately, we are not currently permitted to use hESC in a legal context. We would like the reviewer to understand this situation. To check cell line generality in our method, we conducted differentiation experiments using another iPS cell line (TKDN4m; S5 Fig: mentioned in main text, P16L22-24). 

Lacking human islets or beta-cell lines making hard to judge the correct performance of the techniques.

 As the reviewer pointing out, the comparison with intact islets or MIN6 cells is important to assess our method. We measured the quantitative level of insulin (qRT-PCR) in both human islet cells and our differentiated cells. As shown below, insulin expression level in our differentiated cells is about one-seventh. We believe that the reason for low insulin expression is due to immaturity. Also in many other methods, differentiated cells from iPSC do not necessarily show better performance in insulin expression than human islet, so we would like the reviewer to understand this common problem.

<Bar graph of insulin expression in both differentiated cells and human islet cells>

Lack of flow cytometry to quantify the percentage of insulin-positive cells.

 According to the comment, we attempted to carry out flow cytometry. Unfortunately, we could not examine using 201B7 cells (by the failure of differentiation due to a technical problem). But, we also could conduct similar experiment using TKDN4m. As a result, VC/RS addition increased the ratio of both insulin+GCN- (1.87% to 2.66%), as well as insulin+ (2.71% to 3.27%). We show the result as a supplemental figure (S5C Fig).

Minor concerns:

The quality of the pictures is low.

In the revised version, we uploaded the figures with TIFF format.

In M&M in part of GSIS, they didn’t mention 25 mM glucose.

With reference to the comment, we revised the description of GSIS in M&M section. 

The action the authors should take for re-reviewing the paper:

Confirming the percentage of the insulin-producing cells in the population by flow cytometry.

Confirming the maturity of the differentiated cells by measuring the intracellular Ca+2.

Including the ES-derived beta cells as comparing with iPS-derived beta cells

Including the human islets to confirm the accuracy of the technique and comparing with the maturity of the cells

We thank the reviewer to summarize the point to do. We carried out (1) flowcytometry, (2) Ca2+flux assay, (3) experiment with other iPS cell lines. We also mentioned comparison with human islets. Please check each point-to-point response for detail.

Reviewer #2: The authors reported a method for differentiating insulin-producing cells from hiPSCs and improved glucose-stimulated insulin secretion by adding vitamin C and RepSox as well as by 3D culture.

The weak point of this paper is that the authors use only one cell line as hiPSCs for their experiments. It is difficult to say whether this protocol may be applicable to other hiPSCs unless they show similar data using more than two different cell lines, because hiPSCs have different characteristics.

According to the comments, we performed same experiment with another cell line (TKDN4m). The differentiation efficiency in this cell line was confirmed to be improved by vitamin C and RepSox addition (S5 Fig; mentioned in the main text, P16L22-24). In addition, we obtained insulin positive cells with same protocol using other cell line (like CiRA clinical lines), although we did not compare +/- vitamin C and RepSox. 

Although the authors concluded that they could improve glucose-stimulated insulin secretion, I do not think the supporting data are enough to say so. According to the Figure 5C, C-peptide secretion decreased at 25 mM glucose concentration compared with the level at 2.5mM, which indicates the lack of glucose responsiveness. Moreover, the description of glucose-stimulated C-peptide secretion assays in M&M is not accurate.

Thanks for the reviewer’s comment. We agree that our differentiated cells showed inefficient activity to secrete insulin. To examine the maturity with another methods, we performed Ca2+ flux assay. The result showed that low and high glucose stimulation clearly increased Ca2+ level, especially in the cells differentiated with vitamin C and RepSox, suggesting that the differentiated cells at least possess the ability to respond to glucose (including high glucose challenge). Previous report showed that immature cells secrete low level of insulin continuously (Bonner-Weir and Aguayo-Mazzucato, Nature 535: 365-366 (2016)), thus we currently believe that the inefficiency of insulin secretion response to high glucose will be due to immaturity of the cells.

Also, according to the comment, we revised description of C-peptide assay in M&M section.

I agree with the authors about the necessity of cost reduction for cell therapy. Many researchers tried to induce functional islets by imitating the developmental process for pancreatic islets. It is important to show that the quality of induced βcells did not change despite the simplified method. It would be more convincing if the authors present comparison data with other reported methods to say “ This simplified method will contribute to realizing transplantation therapy of βcells using iPSCs.”

We greatly appreciate that the reviewer understands the importance of cost reduction for the regenerative therapy. We also understand that it is necessary to show the comparison with other methods. According to the comment, we carried out differentiation using one of other protocols (Yabe et al., 2017). As described above, our results indicated that the ratio of insulin positive cells as well as insulin expression in differentiated cells by our method were larger than other method, whereas Maf-A expression in our differentiated cells was lower than other method. This data is shown in S3 Fig.

---

## [Decision Letter · Decision Letter 1]

25 Jun 2021

A simple method of hiPSCs differentiation into insulin-producing cells is improved with vitamin C and RepSox

PONE-D-21-09954R1

Dear Dr. Michiue,

We’re pleased to inform you that your manuscript has been judged scientifically suitable for publication and will be formally accepted for publication once it meets all outstanding technical requirements.

Kind regards,

Makoto Kanzaki, Ph.D.

Academic Editor

PLOS ONE

Additional Editor Comments (optional):

Your revised manuscript was sent to original referees, but one reviewer unfortunately could not review the revised one. After careful reading by myself, reviewer #2 and I found that the authors properly addressed all the concerns raised by both of reviewers and that thus the revised manuscript has been improved. So I made this decision avoiding further delay by another round of reviewing with new reviewer(s). 

Reviewers' comments:

Reviewer's Responses to Questions

**Comments to the Author**

1. If the authors have adequately addressed your comments raised in a previous round of review and you feel that this manuscript is now acceptable for publication, you may indicate that here to bypass the “Comments to the Author” section, enter your conflict of interest statement in the “Confidential to Editor” section, and submit your "Accept" recommendation.

Reviewer #2: All comments have been addressed

2. Is the manuscript technically sound, and do the data support the conclusions?

Reviewer #2: (No Response)

3. Has the statistical analysis been performed appropriately and rigorously? 

Reviewer #2: (No Response)

4. Have the authors made all data underlying the findings in their manuscript fully available?

Reviewer #2: (No Response)

5. Is the manuscript presented in an intelligible fashion and written in standard English?

Reviewer #2: (No Response)

6. Review Comments to the Author

Reviewer #2: (No Response)

7. PLOS authors have the option to publish the peer review history of their article (what does this mean?). If published, this will include your full peer review and any attached files.

Reviewer #2: No

---

## [Editor Report · Acceptance letter]

30 Jun 2021

PONE-D-21-09954R1 

A simple method of hiPSCs differentiation into insulin-producing cells is improved with vitamin C and RepSox 

Dear Dr. Michiue:

I'm pleased to inform you that your manuscript has been deemed suitable for publication in PLOS ONE. Congratulations! Your manuscript is now with our production department. 

Kind regards, 

on behalf of

Dr. Makoto Kanzaki 

Academic Editor

PLOS ONE